# The First Miniature, Small Foliose, Brown *Xanthoparmelia* in the Northern Hemisphere

**DOI:** 10.3390/jof10090603

**Published:** 2024-08-25

**Authors:** Guillermo Amo de Paz, Pradeep K. Divakar, Ana Crespo, Helge Thorsten Lumbsch, Víctor J. Rico

**Affiliations:** 1Departamento de Farmacología, Farmacognosia y Botánica (U.D. Botánica), Facultad de Farmacia, Universidad Complutense, Plaza de Ramón y Cajal s/n, 28040 Madrid, Spain; pdivakar@farm.ucm.es (P.K.D.); amcrespo@ucm.es (A.C.); rico@ucm.es (V.J.R.); 2Collections, Conservation and Research, The Field Museum, 1400 S. Lake Shore Drive, Chicago, IL 60605, USA; tlumbsch@fieldmuseum.org

**Keywords:** *Parmeliaceae*, lichens, evolution, taxonomy, biogeographic disjunction

## Abstract

The genus *Xanthoparmelia* includes several subcrustose, squamulose, small foliose, and small subfruticose species, primarily in the Southern Hemisphere. Here, we report on the first small foliose species lacking usnic acid in the genus occurring in the Holarctic. The species has been previously known as *Lecanora olivascens* Nyl., but subsequent studies of the morphology, secondary chemistry, and molecular data of the nuITS rDNA indicate that this species instead belongs to *Xanthoparmelia*. Consequently, the new combination *Xanthoparmelia olivascens* (Nyl.) V.J. Rico and G. Amo is proposed, and an epitype is designated here. We discuss the unique presence of a subcrustose *Xanthoparmelia* species lacking cortical usnic acid in the Northern Hemisphere. This species fits phylogenetically into a clade that was previously only known from the Southern Hemisphere, and hence represents another example of N-S disjunction in lichenized fungi.

## 1. Introduction

The diversity of fungi and their estimated huge numbers is one of the great questions of evolutionary biology [1]. As has been frequently noted, *Parmeliaceae* is one of the largest families of lichenized fungi, encompassing a major clade primarily composed of foliose to fruticose species known as the parmelioid clade, with approximately 1500 described species [2,3]. Within the parmelioid lichens, nine clades have been identified [4]. Of these, one remarkable monophyletic group is the *Xanthoparmelia* clade, which includes the single genus *Xanthoparmelia* (Vain.) Hale. This is one of the most speciose genera of lichen-forming fungi. Noteworthy, only two genera are known to be ultra-diverse (>500 species) in lichen-forming fungi [5]; one of these is *Xanthoparmelia*. This genus includes over 800 described species predominantly found in arid to semiarid regions, where they thrive on siliceous rocks or soil. Contrarily to the high diversity in Australia, South Africa, and North and South America, fewer species are known from Asia and Europe [6]. *Xanthoparmelia* species are characterized by their *Xanthoparmelia*-type cell wall polysaccharides and arachiform vacuolar bodies in their ascospores [7,8,9]. However, phenotypic variations are found in growth forms, including foliose, fruticose, and subcrustose, and thallus color, including a yellow, grey, and brown upper surface. The latter is due to the presence of usnic acid, atranorin, and melanin substances, respectively, in the upper cortex. Brown subcrustose species are rare in *Xanthoparmelia* and, so far, known from the Southern Hemisphere [10]. 

In the monophyletic and highly diverse genus *Xanthoparmelia*, no robust main clades were identified, which may reflect rapid adaptive radiation following a shift towards drier habitats and a switch to rocks and soils at the base of the *Xanthoparmelia* clade [11,12]. These habitat shifts likely gave rise to various morphological adaptations, including growth forms, attachment structures (e.g., rhizines), apothecia morphology, ascospores, pycnidia, and more, the significance of which has been previously overestimated in classification efforts. The genus *Xanthoparmelia* is a clear example of how phylogenies based on evolutionary DNA markers have been used to reinterpret morphological characters and their evolution and re-evaluate the systematics of the group [7,9,13,14,15]. The phylogenetic multi-locus reconstructions within *Xanthoparmelia* elucidate evolutionary relationships, estimate divergence times, reconstruct biogeographical histories, and hypothesized an African origin during the Miocene [16,17,18]. Moreover, the latter study identifies several monophyletic clusters within *Xanthoparmelia* based on their geographies, such as Holarctic, several South African and Kenya+South African clades, two Australian clades, three Kenyan clades, several other clades specific to taxonomic groups, and a Southern Hemisphere clade. The phenotypic diversity of growth forms observed in different *Xanthoparmelia* clades suggests that these forms may have evolved independently multiple times within the genus. However, the foliose growth form has been reconstructed as an ancestral state for the *Xanthoparmelia* clade [12]. This phenotypic diversity is particularly noteworthy within the “Southern Hemisphere” clade in [18] or in Clade VIII in [19], which incorporates miniature parmelioid species with small laciniate prostrate, small foliose rhizinate, subcrustose (needed of a careful examination), or squamulose peltate thalli. That is, small-scale thalli with a maximum of a few centimeters of development and a close relation with the substrate. What we call miniature species are found in several branches of the genus *Xanthoparmelia* in [18], such as some species of the synonymized genus *Karoowia* [14] or the well-known *X. mougeotii*. They are in the “Southern Hemisphere” clade [18] and in Clade VIII [19], where all the species treated have this miniature morphology.

Over the last couple of years, some of us collected a saxicolous, silicicolous, subcrustose, or small foliose brown lichen tentatively identified as *Xanthoparmelia* in Spain. The material was analyzed, and the possibility of it being a new species was considered. Simultaneously, research on the genus *Protoparmelia* was conducted, involving an analysis of the type specimens of certain names, including *Lecanora olivascens* Nyl. Curiously, the type material of *L. olivascens* was found to be conspecific with our *Xanthoparmelia* sample. The taxonomic placement of *Lecanora olivascens* was unclear since it was described in [20]: it was classified into six different genera over time, most recently in *Protoparmelia* (Llimona in [21]). Studies have repeatedly shown that molecular phylogenies are powerful to address taxonomic issues and the classification of species of lichen-forming fungi and fungi in general (see e.g., [4,22,23,24,25,26,27]). Hence we performed a molecular phylogenetic analysis of *Lecanora olivascens*, including a sample from the type locality, to address the phylogenetic relationships of this species. In addition, detailed morphological, anatomical, and chemical analyses were conducted.

## 2. Materials and Methods

### 2.1. Morphological, Anatomical, and Chemical Examinations

For morphological and chemical studies, we examined eight specimens of the species from the MAF-Lich. and H-NYL herbaria, and eight specimens of additional species were used for comparison from MAF-Lich (Table 1). Morphological and anatomical examinations were performed under a Nikon SMZ-1500 stereomicroscope (Tokyo, Japan) and a Nikon Eclipse-80i microscope (Tokyo, Japan), with bright field and DIC images. Pictures were taken with a Nikon DS-Ri2 digital camera (Tokyo, Japan) coupled to the microscope and the stereomicroscope. Habit pictures were taken with a Fujifilm X-T1 digital camera (Ebina-shi, Japan) paired with a Zeiss Touit f2.8 Macro lens (Jena, Germany), and all were coupled to a Kaiser Reproständer RS1 5507 stand with a Kaiser LED-Beleuchtungseinrichtungen illumination (Schalksmühle, Germany). To combine successive focal image levels, we used the free Combine ZP software v.1.0 from Alan Hadley (downloaded in 2012 but currently not available). Areoles, lobes, and apothecia sizes were determined with a Leitz-Wetzlar 8× scale magnifying lens (Wetzlar, Germany). Observations and measurements of anatomy, ascospores, and conidia were made using thin hand sections mounted in water. Extreme values have been noted between brackets when they represented no more than 5% of the readings. When possible, at least 30 spores and conidia from different thalli were measured with a 100× magnification objective, and only developed spores outside the asci were measured. Current mycological terminology is used following [28]. Spot tests (K, C, I, and PD) and thin-layer chromatography (TLC) were carried out following [29]. We used TLC solvent system C (200 mL of toluene/30 mL of acetic acid) and silica gel 60 F254 aluminum sheets (Merck, Darmstadt, Germany) for the chemical analysis.

### 2.2. DNA Extraction, PCR Amplification, Sequencing, and Phylogenetic Analyses

Two *Xanthoparmelia* specimens were used to generate new sequences: MAF-Lich. 25204 (nuITS GenBank acc. no. PP968419) and MAF-Lich. 24758 (nuITS GenBank acc. no. PP968420). In the megablast search, the ITS sequence of *Lecanora olivascens* showed the highest similarity with *Xanthoparmelia* sequences. Therefore, to elucidate the systematic placement at the genus level of *Lecanora olivascens* specimens, we exclusively employed the nuITS marker and incorporated it into a larger data matrix of 127 specimens that encompasses species of the major clades within the genus *Xanthoparmelia*, as outlined in [18,19].

We performed secondary metabolite extractions from selected lobes for TLC analyses using acetone (Panreac, Castellar del Vallès, Spain). Subsequently, the same lobes were mechanically disrupted using sterile glass pestles in liquid nitrogen and total DNA was extracted from the homogenized samples using the Speedtools plant DNA Extraction Kit (Biotools, Madrid, Spain) by following the manufacturer’s protocol. We used ITS1-LM [30] and ITS-2KL [31] primers for PCR amplification. We used a reaction mixture for amplification containing 5 μL of DNA extraction (second elution), 12.5 μL of DNA Amplitools Master Mix (Biotools), 1.5 μL of each primer, and 4.5 μL of H_2_O. PCRs were carried out in an automatic thermocycler (Bioer Xpcycler, Hangzhou, China). PCRs were performed with the following steps: 1.—initial denaturation at 94 °C for 5 min; 2.—35 cycles of 94 °C for 1 min, 56 °C for 1 min, and 72 °C for 1.5 min; and 3.—a final extension at 72 °C for 5 min. PCR products were cleaned using a DNA purification kit (Biotools). The PCR products were sequenced using the ABI Prism Dye Terminator Cycle Sequencing Ready reaction kit using the same primers (Applied Biosystems, Foster City, CA, USA) and were electrophoresed on a 3730 DNA analyzer (Applied Biosystems).

New sequence fragments were assembled and edited with BioEdit ver. 7.0.9 [32]. We employed two matrices to elucidate the phylogenetic position of the specimens. First, we used a matrix with 127 OTUs representing all the major clades within the genus *Xanthoparmelia* and major clades in parmelioid lichens (as an outgroup). These clades within the genus *Xanthoparmelia* were selected from previous studies [18,19]. Sequences were taken from data sampling of three previous studies [16,17,18] and completed with some sequences downloaded from GenBank. Sequences of matrix 1 (127 OTUs) with the GenBank acc. nos. are listed in Appendix A. In the following step, we used a second matrix with 22 OTUs of the clade “Southern Hemisphere” from [18] and “Clade VIII” [19]. We performed independent alignments for each matrix using the program MAFFT v.7 [33] with the FFT-NS-i alignment algorithm and the remaining parameters were the default values. 

We used maximum likelihood (ML) and Bayesian analyses (BAs) for the phylogenetic analyses. ML analyses were performed in RaxML-HPC2 on XSEDE v.8.2.10 [34] on Cipres Science Gateway (http://www.phylo.org) [35] with the GTR+G+I substitution model and 1000 bootstrap pseudoreplicates. BAs were performed with the same substitution model in MrBayes v. 3.2.6 [36,37]. For BAs, we used two parallel Markov chain Monte Carlo (MCMC) runs with 8,000,000 generations in each run and sampled every 1000 steps. We discarded the first 3000 trees as burn-in, and with the rest, we used a 50% majority-rule consensus tree. The convergence parameters were close to 1.0 in all the cases. We used Fig Tree v.1.4.0 [38] to visualize the tree files.

### 2.3. Cell Wall Polysaccharide Analysis

The type of lichenan in the hyphal cell wall was determined with the histochemical method following [39,40]. The reagents were prepared following [29]. We follow the same steps as explained in [15] using *Xanthoparmelia mougeotii* and *Parmotrema reticulatum* as control taxa (specimens listed in Table 1). 

## 3. Results and Discussion

### 3.1. Phylogeny and Morpho-Chemical Studies

We show the results of the phylogenetic analyses using maximum likelihood (ML) and Bayesian analyses (BAs) with the two data matrices (Figure 1 and Figure 2). In both cases, ML analyses and BAs were congruent. Figure 1 shows the results with the matrix of 127 taxa. In this analysis, the two specimens (one from Spain and another from France, the type locality) of *Lecanora olivascens* grouped within the genus *Xanthoparmelia*. The clades presented in Figure 1 are congruent with the major phylogenies of the genus *Xanthoparmelia* found previously [18,19]. In *Xanthoparmelia*, samples of *L. olivascens* clustered within the so-called “Southern Hemisphere” clade [18] or Clade VIII [19]. The circumscription of Clade VIII in [19] is a little broader, grouping its sister clade formed by specimens of *Xanthoparmelia subruginosa*, but the topologies are also congruent. 

This phylogenetic result is congruent with the results of our morpho-chemical studies, i.e., the presence of an arachiform vacuolar body in the ascopore (Figure 3g) and the presence of the xantholichenan-type cell wall polysaccharide. The shape of the vacuolar body and the type of the cell wall polysaccharide are the two taxonomic characters to recognize the *Xanthoparmelia*-clade [7,9]. Consequently, a new combination of that species into *Xanthoparmelia* is proposed below.

Figure 2 shows the result of the phylogenetic analysis (ML and BA) of the data matrix 2 (22 taxa), including the samples grouped in Clade VIII sensu [19]. The two samples of *Xanthoparmelia olivascens* MAF-Lich. 25204 (France, nuITS GenBank acc. no. PP968419) and MAF-Lich. 24758 (Spain, nuITS GenBank acc. no. PP968420) grouped in the same supported clade inside Clade VIII. In this clade, we can find a group of foliose and subcrustose species of *Xanthoparmelia* with usnic acid in the cortex from South Africa and Kenya. In addition, there is an unsupported clade without usnic acid in the cortex. This latter clade, including *X. olivascens*, also contains other species from South Africa and South America. Although subcrustose species mainly cluster in this clade, there are also species with squamulose peltate morphologies (*X. peltata*) or species with a laciniate prostrate growth form (*X. azaniensis* and *X. ovealmbornii*), which were previously considered as independent genera *Omphalodiella* [14,41] and *Almbornia* [13,42], respectively. To our knowledge, *Xanthoparmelia olivascens* is the only Holarctic species in Clade VIII dominated by species from South Africa, South America, and East Africa. 

### 3.2. Taxonomy

***Xanthoparmelia olivascens*** (Nyl.) V. J. Rico and G. Amo **comb. nov.**
Figure 2 and Figure 3.

≡ *Lecanora olivascens* Nyl., Bull. Soc. Linnéenne Normandie, sér. 2, 6: 263, 1872 (Basionym).

≡ *Lecanora badia* var. *olivascens* (Nyl.) Boistel, Nouv. Fl. Lich., II partie (partie scientifique): 142, 1903.

≡ *Squamaria olivascens* (Nyl.) H. Olivier, Mém. Soc. Sci. Nat. Cherbourg 37: 61, 1909.

≡ *Solenopsora olivascens* (Nyl.) M. Choisy and Werner, Bull. Soc. Hist. Nat. Afr. N. 22: 11, 1931.

≡ *Placolecanora olivascens* (Nyl.) Räsänen *ex* B. de Lesd., Rev. Bryol. Lichénol. 18: 75, 1949.

≡ *Placodium olivascens* (Nyl.) Motyka, Porosty (Lichenes). 4, Rodzina *Lecanoraceae*. *Placodium*, *Squamarina*, *Harpidium*, *Trapelina*, *Mosigia* (Lublin): 84, 1996.

≡ *Protoparmelia olivascens* (Nyl.) Llimona *ex* J.-M. Sussey, Bull. Inf. Ass. Française Lichénol. 36(1): 57, 2011.

**Type:** France, Pyrénées-Orientales, “Schistes argileux de Força-Réal, *W. Nylander* s.n., [16-VII-]1872” (**Holotype:** H-NYL 25742 = H9506918!; with norstictic and stictic acids + unknown substances according to F.J. Walker 1980 in a label); France, Occitanie, Pyrénées-Orientales, Perpignan, Millas, Força-Réal, SSE steep slope, 42°43′28.9′′ N, 02°42′08.6′′ E, 366 m, *Quercus ilex–Q. coccifera* forest with profuse bushes on slates and schists with quartz intrusions, 12-VI-2023: *G. Amo de Paz and V. J. Rico 5190/1* (**epitype designated here:** MAF-Lich. 25204!), GenBank: ITS = PP968419, MycoBank: MBT 10021536; ibid.: *G. Amo de Paz and V. J. Rico 5190/2, 5191, 5192* (**isoepitypes:** MAF-Lich. 25205, 25206, 25207!).

**MycoBank**: MB 855007.

**Description:** Thallus saxicolous silicicolous, small foliose (subcrustose), ± areolate in the central part and radially lobate in the margins, appressed and adnate to the rock, 0.2–3.0 cm in diameter; fertile but without soredia, isidia, or pseudocyphellae. Lobes 0.2–2.5 mm long × 0.15–1.2 mm wide, linear elongate to ± deeply sinuated, dichotomously to palmately branched at the ends, the longest with transverse fractures that originate the areoles, mainly contiguous, convex but sometimes ± plane at lobe ends, ends rounded, surface smooth, dull or slightly shiny at the ends, lateral margin of the lobes gently descending to the lower cortex and darker than the surface to black, loose to rhizinate or attached to the rock by tufts of lower cortex rhizoidal hyphae. Inward areoles 0.3–1.5 mm diameter, polygonal to rounded or elongate, sometimes fissured, convex, surface smooth to rugose by apothecia primordia or by regeneration also in new lobes, lateral margin of the lobes mainly angled or gently descending to the lower cortex, marked, dark brown to black, adnate and appressed to rarely loose, areoles seems to be originated by lobules fragmentation or arise from divisions of the lobes during their growth. Upper surface dark reddish-brown to dark greyish-brown or brown-black, sometimes paler olivaceous to yellowish in the lobe tips. Lower surface mainly black, brown, light brown to beige, dull, flat to somewhat irregular, free or with simple, short, sparse, concolorous to dark rhizines in lobed loose areas, or directly fixed to the substratum by tufts of rhizoidal hyphae from the lower cortex, especially in central areolate areas. Thallus heteromerous. Upper cortex 7–37 μm tall, reddish-brown in the upper 2–3 cell rows, distinctly limited and paraplectenchymatous, K+ yellowish under microscope, with a gelatinous epicortex. Algal layer up to 90 μm tall, irregular and continuous to rarely discontinuous, K-, with *Trebouxia*-type algal cells, up to 18 μm in diameter. Medulla hyaline or greyish (under stereomicroscope) or greyish (under microscope), variable in thickness, mainly with irregularly branched, intermixed, pachydermatous hyphae covered with greyish crystals (microscope), K+ red generating acicular crystals. Lower cortex 14–60 μm tall, or probably more in central areoles, light brown, reddish-brown to dark brown or black, distinctly delimited and paraplectenchymatous, with 3 or more cell rows, free or attached to the substrate by simple, short dark rhizines (this in the lobes) or by tufts of light brown to beige strands of rhizoidal hyphae, with ±short, parallel, conglutinated and gelatinous cells (seen when the lobes/areoles are removed, part of the lower cortex remains attached to the rock), K+ yellowish under microscope.

Ascomata apothecia, common, lecanorine, immersed when young becoming adnate or sessile, sometimes slightly constricted at the base when adult, rounded to irregular by pressure, up to 1.1 mm in diameter, margin entire, even to irregular, 1 to several per areole and mainly at thallus center. Disc black to dark brown, dull, flat to slightly concave or slightly convex. Thalline exciple persistent, concolorous with thallus, smooth and dull, forming an entire to slightly crenate margin (stereomicroscope). Proper exciple cupulate, coherent, hyaline, brown in the exposed sometimes slightly flabellate margins where, under the microscope, it is clearly located between the hymenium and the thalline exciple, 30–60 μm thick in the center of apothecia. Hymenium hyaline to ochraceous, coherent, 40–60 µm tall; epihymenium brown to reddish-brown, up to 20 µm tall, upper with an amorphous reddish gelatinous matrix; subhymenium hyaline to slightly ochraceous, reduced to a narrow band. Paraphyses coherent in water, branched and anastomosed, pachydermatous, apices thickened (up to 7 μm in K, and up to 8 µm wide in water), with brown light caps and a hyaline to brown mucilaginous hood. Asci clavate, claviform, eight spored, tholus and surrounding mucilage amyloid (excluding the axial mass), *Lecanora* type. Ascospores hyaline to slightly yellow-orange, simple, 12–17.5 (–23) × 4.5–7.5 μm (n = 80), narrowly ellipsoidal to oval or cylindrical (l:b = (1.7–) 2.0–3.0 (–3.8), l:b average = 2.4), wall 0.5–1.0 μm wide, with arachiform vacuolar body (Figure 3g).

Pycnidia frequent, immersed, laminal, punctiform, globose to oblong, up to 120 μm wide, wall hyaline, ostiole tissue with brown to black pigmented walls. Conidiophores similar to type V with shorter ramifications [43], and intermediate to those in *Protoparmelia* and *Pleurosticta* in [44]. Conidia simple, hyaline, narrowly fusiform, straight, 7–12 × 1–2 µm (n = 40).

**Chemistry:** Cortex HNO+ blue-green; medulla K+ yellow turning red and producing large, acicular crystals, C-, PD+ yellow, I-. Constituents (TLC): norstictic acid (major), hyposalazinic acid (minor), connorstictic acid (minor), consalazinic acid (minor), subnorstictic acid (traces), and stictic acid (traces).

**Habitat and distribution:** *Xanthoparmelia olivascens* is currently known from its type locality in France (Força-Réal) and from central Spain (San Martín de Valdeiglesias, Madrid). These locations have a NW, W to SW exposure and feature typical Mediterranean evergreen *Quercus* forests and garrigue formations on siliceous substrata. In the type locality, it grows on slates and schists with quartz intrusions, often accompanied by *Acarospora* spp., *Aspicilia cinerea*, *Buellia atrocinerella*, *B. stellulata*, *Monerolechia badia*, *Rhizocarpon* spp., *Usnochroma carphinea*, and *Xanthoparmelia verruculifera*. At times, it may be overgrown by *M. badia* and *X. verruculifera*, and especially in its marginal areas, it can be mistaken for small lobes lacking isidia of *X. verruculifera*. In the Madrid locality, where it is relatively common, *Xanthoparmelia olivascens* grows on horizontal to sloped granite rock formations, which are highly exposed to sun, rain, and winds, primarily in northwest to southwest orientations, corresponding to the prevailing humid winds. In addition to known Mediterranean species, it has been collected alongside *Acarospora fuscata*, *Candelariella vitellina*, *Circinaria hoffmanniana*, *Dimelaena oreina*, *Miriquidica deusta*, *Pyrenopsis triptococca*, *Rhizocarpon geographicum* agg., and *Xanthoparmelia tinctina*. Frequently, it intermingles with or serves as a substrate for *A. fuscata* and *P. triptococca*.

Historical literature records of *Xanthoparmelia olivascens*, or some of its synonyms, span across various European regions, including France, Corsica, Bulgaria, Greece, Spain, and the Canary Islands [45,46,47,48,49,50,51]. However, the taxonomic accuracy of these records remains uncertain and warrants a comprehensive examination. Given the potential existence of different *Protoparmelia* species within these records, at least some of those may not belong to *X. olivascens* but similar *Protoparmelia* species.

No lichenicolous fungi were observed in the studied specimens.

**Additional specimens examined: France**, Pyrénées-Orientales, Força-Réal: 1-VI-1884, *W. Nylander* (H-NYL p.m. 3299 = H9234912); id.: 300 m, 1-VI-1884, *W. Nylander* s.n. (H-NYL 25741 = H9234913). **Spain**, Madrid, San Martín de Valdeiglesias, desembocadura del río Cofio en el embalse de San Juan, cerro de El Yelmo, 664 m, 40°23′21′′ N, 04°18′50′′ W, sobre granito: 6-VI-2013, *V.J. Rico 4605/1* (MAF-Lich. 24759); id.: 21-II-2014, *V.J. Rico 4616/1*, *4616/2*, *4616/3*, *4616/4*, *A. Crespo and C. Ruibal* (MAF-Lich. 24760, 24761, 24762, 24763).

**Comments:** In Nylander’s original description of *Lecanora olivascens* in [20], no type specimen was explicitly indicated. However, within the collections held at H-NYL, three original Nylander specimens labeled as *Lecanora olivascens* are preserved. It is noteworthy that two of these specimens (H-NYL 25741 = H9234913 and H-NYL p.m. 3299 = H9234912) were mistakenly designated as lectotype and type by J. Motyka in 1965 and F.J. Walker in 1980, respectively. Additionally, Llimona in 1979 erroneously ranked one of these specimens (H-NYL p.m. 3299 = H9234912) as an isotype, as also noted B. Ryan in 1992 in a separate tag. Since these two specimens were collected in 1884, which is twelve years after the original description, they cannot be considered as type specimens.

The third folder (H-NYL 25742 = H9506918) contains material collected by Nylander from the original locality at the time of the species’ publication [20]. This folder includes a separate tag, incorrectly labeled as syntype by J. Motyka in 1965. The original label of this third folder contains manuscript notes by Nylander, which clearly correspond to different characters described in the protologue (Figure 4, left panel). It is evident that this material was used for the original species description. As no other original material was located, we consider the H-NYL 25742 = H9506918 specimen of *Lecanora olivascens* as the holotype, following Art. 9.1 of the International Code of Nomenclature for algae, fungi, and plants (ICN, [52]).

Although the holotype material is limited and aged (Figure 4, right panel), we examined it; however, it is physically incomplete, and we did not attempt to study its marginal lobes and the presence or absence of rhizines, its anatomy or to extract DNA or conduct a TLC analysis, which conveys a certain hesitation and uncertainty in the application of the name. It is deductible, nonetheless, that the holotype material taxonomically belongs to *Xanthoparmelia* and comprises a thin, small foliose (subcrustose), brown lichen with lecanorine and flat apothecia. This appearance could also lead to potential confusion with tiny thalli of *Xanthoparmelia pulla* group taxa, with which it occasionally cohabits. TLC analyses performed by F.J. Walker in 1980 indicated the presence of norstictic and stictic acids, accompanied by unidentified substances (as noted in a separate tag). On the other hand, the H-NYL not type specimens mentioned above (H-NYL 25741 = H9234913 and H-NYL p.m. 3299 = H9234912) are conspecific with the holotype and have helped us to characterize the species and include it in the genus *Xanthoparmelia*, since they are larger, are preserved in better condition, and we have been able to morphologically study them in more detail, although the cells are collapsed.

To provide precision in the application of the name and to establish its taxonomy and phylogenetic position within *Xanthoparmelia*, considering Art. 9.9 of the International Code of Nomenclature for algae, fungi, and plants (Shenzhen Code, [52]; San Juan Chapter F, [53]) and following suggestions by [54], we have designated here a morphologically studied, DNA-sequenced, and TLC-analyzed epitype (Figure 3). This epitype aligns morphologically, chemically, and geographically with the holotype. Furthermore, the selected epitype and isoepitypes were collected by us in the original type locality, Força-Real, in Southeast France. As a result, the morphological description, chemical analysis, and molecular data primarily rely on samples from the epitype and isoepitypes collected at the type locality, in addition to the Spanish specimens and the not type Nylander specimens.

The name *Lecanora olivascens* has an intriguing history marked by uncertainties stemming from its original description. In Nylander’s protologue [20], he described this lichen as cracked-areolate, thin, adnate, and dark brown, featuring contiguous lobate margins and lecanorine apothecia. However, he expressed uncertainties about its taxonomic placement. Nylander compared the newly described species to “*Lecanora*” *montagnei* (now recognized as *Protoparmelia montagnei* (Fr.) Poelt and Nimis 1987) and “*L*.” *badia* (currently *Protoparmelia badia* (Hoffm.) Hafellner 1984). He mentioned that he would have categorized the species in the genus *Parmelia* (in the sense of the 19th century) based on its thallus development, which resembled the more placodioid or parmelioid growth form, as well as its subfusiform acicular conidia (referred to as bacilliform in “*Lecanora*” *badia* and related species). However, he noted that macroscopically the apothecia are lecanorine and not cup-shaped, as seen in *Parmelia*. Nylander’s observations led him to conclude that paraphyses and pycnidia in the new species were similar to those in “*Lecanora*” *badia*, “*L*.” *montagnei*, and *Parmelia* species, thus establishing a connection between these groups of taxa, which are now recognized at different subfamily levels in *Parmeliaceae* [24]. With these brief notes, Nylander placed *Lecanora olivascens* within the “*Lecanora*” *badia-montagnei* group.

Furthermore, Nylander reiterated the position of the species in 1891 [55], providing a similar description and discussion to that in 1872. He replaced “parmelioid” thallus with “subparmelioid” thallus in his short discussion, ultimately concluding that *Lecanora olivascens* is a lineage of “*Lecanora*” *badia*. Subsequently, later authors, in a sort of refereed records, likely without direct access to the original material and following Nylander’s conclusion that *Lecanora olivascens* was a crustose, non-parmelioid species, combined it into various non-parmelioid genera (see synonyms). Eventually, it was classified within the genus *Protoparmelia* (Llimona in [21]), although photographs of a *Protoparmelia*, which is part of our study, were included, but not a small-foliose *Xanthoparmelia*. This uncertainty regarding the classification of *Lecanora olivascens* has persisted for about 150 years.

During the 20th century, most authors followed Nylander’s circumscription of our species within the “*Lecanora*” *badia-montagnei* group, with Harmand [56] notably highlighting this placement. Poelt [57] described a group of taxa known as the “*Lecanora-olivascens*-Gruppe” within the lobed *Lecanora* species. This group included our species along with “*Lecanora*” *demissa* (currently *Olegblumia demissa*, placed in *Teloschistaceae*). In his discussion of the group, Poelt [57] partially linked *Lecanora olivascens* with the South African subcrustose brown species *Placolecanora natalensis* (synonymous with *Xanthoparmelia dregeana* according to [58], as *Neofuscelia dregeana*, which requires revision). Eigler [59], in a comprehensive study following Poelt’s groups, suggested that *Lecanora olivascens* should be placed in a separate genus distinct from *Lecanora*. Finally, Ryan and Nash III [60] refer to Poelt’s “*Lecanora olivascens*-Gruppe” as being of uncertain position.

This historical account underscores the importance of meticulous morphological studies in shedding light on the taxonomy and systematic placement of lichenized fungi. Such studies complement molecular phylogenetic analyses and contribute to our understanding of evolutionary traits.

Our data reveal some variability among the studied populations of *Xanthoparmelia olivascens*, both in central Spain and southeastern France. In the southeastern France population, the thallus exhibits a more greyish-brown hue, with rhizines being more frequent in the lobes. Additionally, this population produces a higher proportion of longer ascospores and conidia, falling within the previously described measurements. Conversely, the central Spain population features a more reddish-brown hue on the thallus surface, with rhizines being relatively rare in the lobes. Moreover, it displays a higher proportion of shorter ascospores and conidia. Furthermore, the ITS sequences of the central Spain population exhibit three substitutions, if we compare them with those of the southeastern France population. While it is known that *Xanthoparmelia* includes species with differences comparable to those outlined above (e.g., *Xanthoparmelia pulla* group), this aspect falls beyond the scope of our current contribution. We believe it is prudent to await further material and data from various locations in the Mediterranean region and to gain a better understanding of specific morphological traits (e.g., thallus structure, ascospores, pycnidia, and conidia) before drawing taxonomic conclusions in this complex.

*Xanthoparmelia olivascens* exhibits an appressed and adnate thallus structure characterized by developing a lower cortex, an areolate inner part, and lobate margins, but in small-scale compared with other European brown foliose species of *Xanthoparmelia*. Technically it is classified as a foliose lichen [28], but with miniature thalli, small loosely arranged lobes, and short and simple rhizines that originate from the lower cortex (Figure 3d). Additionally, it features an areolate center firmly anchored to the substrate by tufts or strands of rhizoidal hyphae emerging from the lower cortex (Figure 3c). As was described in other comparable *Xanthoparmelia* species [61], the areoles seem to have been formed by the growth of the lobes, their fragmentation into portions, and, subsequently, by a stronger attachment to the substrate. This attachment strategy is such that when these areoles are removed, a portion of the lower cortex remains affixed to the substrate by tufts or strands of lower cortex rhizoidal hyphae; in this way, it can be confused with a crustose lichen.

In contrast, species of *Protoparmelia* typically possess a clearly crustose thallus structure, which can range from areolate to squamulose. They lack a lower cortex and rhizines, and occasionally have margins that are not clearly lobated [62,63].

These types of miniature foliose lichens with a lower cortex and rhizines are often referred also to as subcrustose (e.g., [14,61]). Similar attachment strategies, involving lobate and rhizinate margins along with rhizoidal strands of lower cortex hyphae in the thallus center, have been observed in other genera within *Parmeliaceae*, such as *Hypogymnia* [64], and in different *Xanthoparmelia* species, such as in *Xanthoparmelia pulla* species group and in *X. mougeotii* (own observations). However, those small lichens are not clearly subcrustose. We introduce the terminology miniature or small-scale foliose and miniature or small-scale fruticose lichens for these types of species in *Xanthoparmelia* since they develop a lower cortex throughout their thallus, from which rhizines emerge, reserving the term subcrustose for those species that are lobed but without a lower cortex and true rhizines throughout the entire thallus. We assume that this type of attachment to the substrate would probably favor the radial growth of the thalli in dry and warm rocky places. We also believe that these terms best fit to the thallus morphology of this type of lichens. It is worth noting that further study and detailed comparisons among lichens with similar features in *Xanthoparmelia* will be necessary to refine our understanding of these unique thallus organizations.

Based on our current knowledge and incorporating the molecular phylogenetic findings, *Xanthoparmelia olivascens* stands out as a unique brown *Xanthoparmelia* species in the Northern Hemisphere characterized by a combination of distinctive traits: (a) thallus morphology—*X. olivascens* is the sole brown species in the Northern Hemisphere that develops miniature small-foliose thalli ([14,15], and this characteristic is also supported by recent regional floras); (b) ascomatal characters—this species produces long ascospores (measuring 12–17.5 (–23) × 4.5–7.5 μm), which are narrowly ellipsoidal to oval or cylindrical, contrasting with the majority of *Xanthoparmelia* species, whose ascospores typically range from 6–13 × 4–7 μm and are mainly ellipsoidal [9,65]; (c) chemistry—*X. olivascens* is characterized by the production of norstictic acid as the major substance, along with hyposalazinic, connorstictic, and consalazinic acids as minor compounds, and this chemical profile forms a distinctive combination among brown *Xanthoparmelia* species in the Northern Hemisphere but is common among *Xanthoparmelia* species in the Southern Hemisphere [10]; and (d) distribution—currently, *X. olivascens* is exclusively known from siliceous rocks in Mediterranean localities within southwestern Europe. Table 2 shows a comparison of morphological, chemical, and distributional characters of brown, small foliose *Xanthoparmelia* species.

## Figures and Tables

**Figure 1 jof-10-00603-f001:**
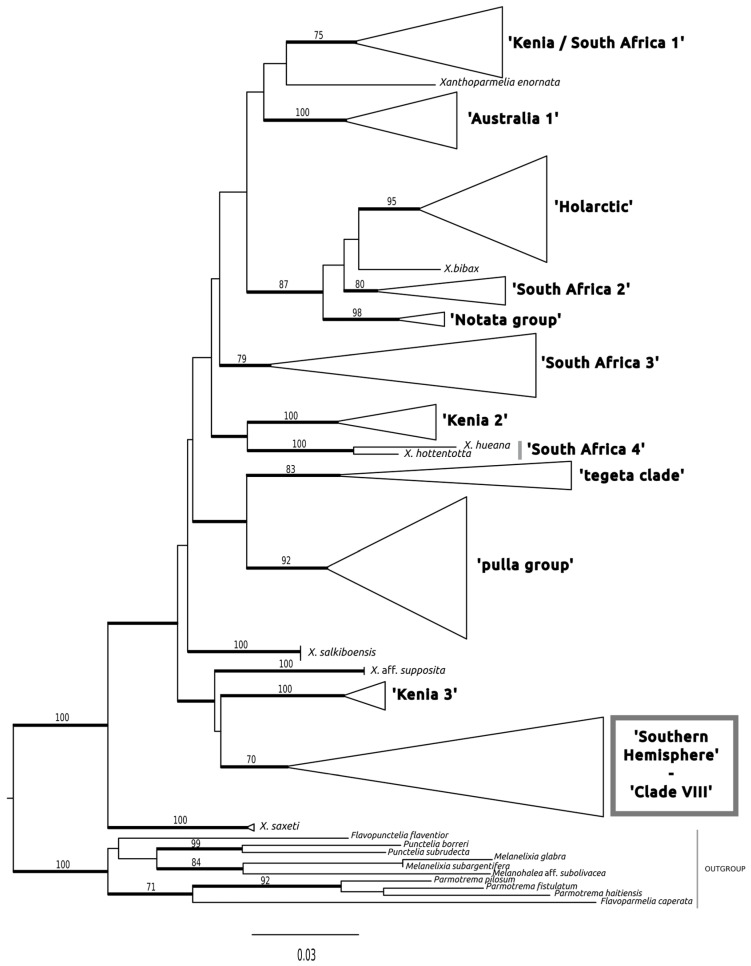
Phylogenetic tree with 127 taxa showing the phylogenetic placement of *Xanthoparmelia olivascens* in the *Xanthoparmelia* clade. Maximum likelihood tree inferred from nuITS sequences. The name of the major clades inside the *Xanthoparmelia* genus follows [18,19] for Clade VIII. Branches with posterior probabilities equal to or above 0.95 are shown in bold and bootstrap values equal to and above 70% under ML are indicated on the branches. *Xanthoparmelia olivascens* appears inside the “Southern Hemisphere—Clade VIII” clade, framed in grey. Tree by G. Amo de Paz.

**Figure 2 jof-10-00603-f002:**
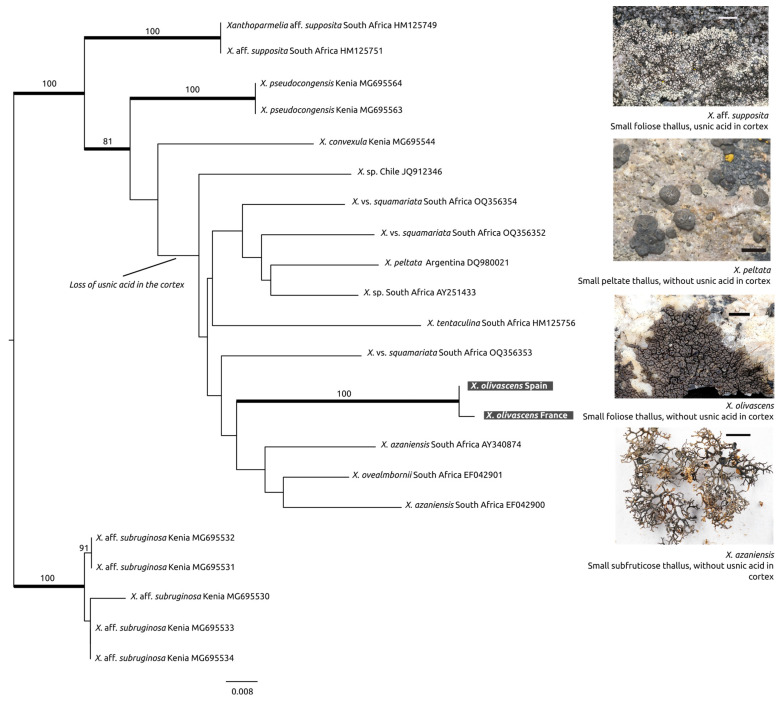
Phylogenetic tree with 22 taxa showing the phylogenetic placement of *Xanthoparmelia olivascens* in the “Southern Hemisphere–Clade VIII” clade inside the *Xanthoparmelia* clade [18,19]. Maximum likelihood tree inferred from nuITS sequences. Branches with posterior probabilities equal to or above 0.95 are shown in bold and bootstrap values equal to and above 70% under ML are indicated on the branches. Images by V.J. Rico, and tree by G. Amo de Paz. Scale bars: 3 mm. Samples photographed: MAF-Lich. 16206 *Xanthoparmelia* aff. *supposita*, UPS-11036a (Lumbsch et al.) *Xanthoparmelia peltata*, MAF-Lich. 24760 *Xanthoparmelia olivascens*, and MAF-Lich. 14269 *Xanthoparmelia azaniensis*.

**Figure 3 jof-10-00603-f003:**
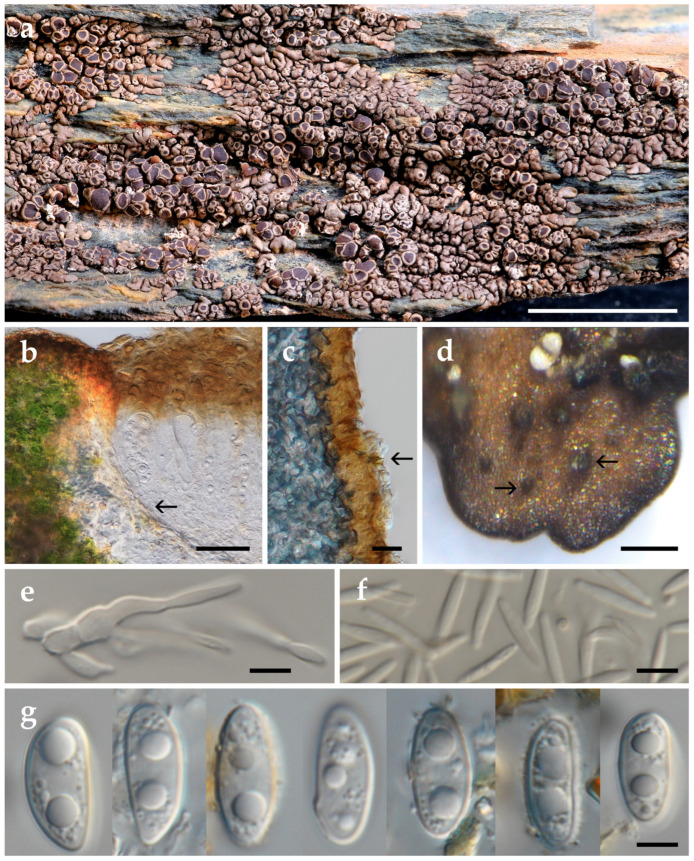
*Xanthoparmelia olivascens*, epitype (MAF-Lich. 25204), except c (MAF-Lich. 24759): (**a**) habit, stacked; (**b**) section of apothecium margin, note the cupular proper excipulum (arrow), DIC, stacked; (**c**) section of thallus areole, showing the medulla (left layer) and the lower cortex with raising tufts of attaching rhizoidal hyphae (arrow), DIC, stacked; (**d**) lower surface with short rhizines (arrows); (**e**) conidiophores, DIC; (**f**) conidia, DIC; (**g**) ascospores, DIC. Images by V.J. Rico. Scale bars: (**a**) = 5 mm; (**b**) = 15 μm; (**c**) = 20 μm; (**d**) = 0.5 mm; and (**e**–**g**) = 5 μm.

**Figure 4 jof-10-00603-f004:**
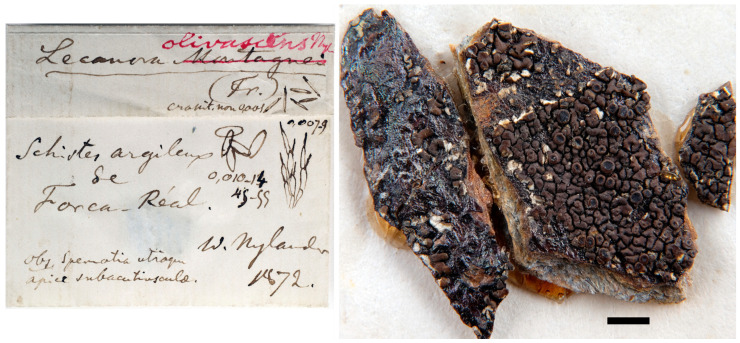
*Lecanora olivascens* Nyl. Holotype (H-NYL 25742 = H9506918): (**left panel**), label; (**right panel**), specimen, stacked. Images by V.J. Rico. Scale bar = 1 mm.

**Table 1 jof-10-00603-t001:** Samples used in the morphological and chemical analyses with the herbarium no.

Herbarium Acc. No.	Locality	Species
H-NYL 25742 = H9506918	Força-Real, Perpignan, France	*Xanthoparmelia olivascens* (type locality)
MAF-Lich. 25204	Força-Real, Perpignan, France	*Xanthoparmelia olivascens* (type locality)
MAF-Lich. 25205	Força-Real, Perpignan, France	*Xanthoparmelia olivascens* (type locality)
MAF-Lich. 25206	Força-Real, Perpignan, France	*Xanthoparmelia olivascens* (type locality)
MAF-Lich. 25207	Força-Real, Perpignan, France	*Xanthoparmelia olivascens* (type locality)
MAF-Lich. 24758	San Martín de Valdeiglesias, Madrid, Spain	*Xanthoparmelia olivascens*
MAF-Lich. 24760	San Martín de Valdeiglesias, Madrid, Spain	*Xanthoparmelia olivascens*
MAF-Lich. 24761	San Martín de Valdeiglesias, Madrid, Spain	*Xanthoparmelia olivascens*
MAF-Lich. 16206	Coegmansilellof, Cape Province, South Africa	*Xanthoparmelia aff. supposita*
MAF-Lich. 17125	Temuco, IX Región, Chile	*Xanthoparmelia* sp.
MAF-Lich. 16445	Alexander Bay, N Cape Province, South Africa	*Xanthoparmelia tentaculina*
MAF-Lich. 14268	Matroosberg, Cape Province, South Africa	*Xanthoparmelia ovealmbornii*
MAF-Lich. 14269	Matroosberg, Cape Province, South Africa	*Xanthoparmelia azaniensis*
MAF-Lich. 21716	Flinders Ranges, South Australia, Australia	*Xanthoparmelia subcrustulosa*
MAF-Lich. 11441	Río de Cárdenas, Zamora, Spain	*Xanthoparmelia mougeotii*
MAF-Lich. 16978	Pico del Inglés, Tenerife, Spain	*Parmotrema reticulatum*

**Table 2 jof-10-00603-t002:** Comparison of morphological, chemical, and distributional characters of selected brown, small foliose *Xanthoparmelia* species. ma = major acid; mi = minor acid; tr = substance in trace amounts.

	*Xanthoparmelia hensseniae*	*X. squamariatella*	*X. subsquamariata*	*X. substygiodes*	*X. patagonica*	*X. olivascens*
Thallus ^1^	>2 cm wide	1–6 cm wide	>4 cm wide	>6 cm wide	>2 cm wide	0.2–3 cm wide
Lobes	sublinear to subirregular, 0.2–0.5 × 0.8 mm wide	contiguous to slightly imbricate, 0.3–0.8 (–1.0) mm wide	flat to weakly convex, sublinear to subirregular, 0.3–1.0 mm wide	discrete and more or less elongate	1–2 × 0.2–0.6 mm wide	linear to palmate, 0.2–2.5 × 0.15–1.2 mm wide
Lower cortex	pale to black	pale to brown	pale	pale, tan to blackish	absent, medulla black pigmented under apothecia	black, brown, light brown to beige
Rhizines	moderately rhizinate	erhizinate or unclearly rhizinate	moderately rhizinate	erhizinate	erhizinate or unclearly rhizinate	rhizinate in lobes, tufts of rhizoidal hyphae in areoles
Ascospores	ellipsoidal, 12–13 × 4.5–6 µm	globose to ellipsoidal, 5–7 × 3.5–5.5 µm	ellipsoidal, 7–9 × 4–6 µm	globose, c. 5–5.5 μm wide	globose to ellipsoidal, 3.5–7 × 3.5–6 µm	narrowly ellipsoidal to cylindrical, 12–17.5 (–23) × 4.5–7.5 μm
Conidia	bacilliform to bifusiform, 8–10 (–14) × 1 µm.	bacilliform, 5–6 × 1 µm	bacilliform to weakly bifusiform, 6–7 × 1 µm	fusiform to bifusiform, c. 6–8 × 1 μm	filiform to slightly bifusiform, 9–16 × 0.8 μm	bacilliform to narrowly fusiform, 7–8 × 1–1.5 µm
Chemistry ^2^	connorstictic (mi), cryptostictic (tr)	connorstictic (mi), hyposalazinic (mi)	caperatic (ma), connorstictic, norcaperatic (mi), hyposalazinic (tr)	hyposalazinic (mi), connorstictic (tr)	connorstictic, cryptostictic (mi), hyposalazinic (tr)	connorstictic, hyposalazinic, consalazinic (mi), subnorstictic, stictic (tr)
Distribution	South Africa	Australia, New Zealand	South Africa	South Africa	South America	southwestern Europe

^1^ All species with a miniature, small foliose thallus. ^2^ All species with norstictic acid as the major substance.

## Data Availability

The sequences presented in this study are openly available at https://www.ncbi.nlm.nih.gov/ (accessed on 20 July 2024) (see Appendix A for the accession numbers). The new combination and the selected epitype were registered in MycoBank (http://www.mycobank.org/ (accessed on 20 July 2024)). The materials were collected in France according to the Nagoya protocol IRCC number ABSCH-IRCC-FR-261975-1 and were deposited in the MAF-Lich. Herbarium.

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
