# Peer review of "The First Miniature, Small Foliose, Brown Xanthoparmelia in the Northern Hemisphere"

_jof, 2024, doi:10.3390/jof10090603_

Round 1

Reviewer 1 Report

Dear Editor,

I carefully read the ms "The first miniature, small-foliose, brown Xanthoparmelia in the northern Hemisphere" submitted by Amo and co-authors for possible publication on the Journal of Fungi, special issue Lichen Forming Fungi—in Honour of Prof. Ana Rosa Burgaz

It was a pleasure to read this very interesting ms thar refers to an important discovery in lichen taxonomy. In my opinion it can be published in the present form. Congratulations to the authors.

I have not comments

Author Response

Thank you very much for taking the time to review this manuscript

Reviewer 2 Report

Dear Editor and Authors,

in the manuscript, a new combination for small subcrustose lichen Xanthoparmelia olivascens (Nyl.) V.J. Rico & G. Amo is proposed and it is put in connection with the previously known species Lecanora olivascens Nyl. Based on newly collected material from Spain and France (also from type locality of L. olivascens), the authors studied morphological, chemical and molecular characteristics of the material and designed an epitype there. New and interesting issue is its distribution - the species fits phylogenetically into a clade that has been previously only known from the Southern Hemisphere. In the manuscript, a first record from Northern Hemisphere is mentioned.

The manuscript is well-written and clear (I appreciate missing typos). The data are sufficiently supported by the analyses selected.  The section Results and Discussion is robust and coherent.

The literature is selected appropriately. The higher number of self-citations is because this paper follows-up previous studies on Parmeliaceae, a lichen group extensively studied by authors.

I recommend to accept the paper for publication.

Please note that the citations in the text are written as full citation and should be numbered and placed in square brackets before publishing (see recommendations for journal style).

Author Response

Thank you very much for taking the time to review this manuscript. We will include the citations in the text numbered and placed in square brackets, following the recommendations for journal style in the revised manuscript.

Reviewer 3 Report

The paper is about the recombination of Lecanora olivascens in the genus Xanthoparmelia based on morphological, chemical and molecular genetic studies. It was also shown that this is the first small foliose species found in the Northern Hemisphere among Xanthoparmelias lacking usnic acid. It points out the morphological diversity of Xanthoparmelia, mainly in the small lobed forms.

All detailed comments are given directly in the manuscript file (pdf).

Author Response

Thank you very much for taking the time to review this manuscript and thank you very much for your comments and corrections. Your revision has improved the article. We will provide a point-by-point response, here, on the main issues, and also in the revised manuscript where we saved all the changes with the option "track changes".

Comments 1: [Title, abstract & results]

Response 1: We have corrected title and abstract (and the whole manuscript) following your proposals to be more precise and clearer in the use of morphological terminology. The abstract and summary were confusing between the different versions of the manuscript but we think they now match well and we do not lose clarity at the beginning of the article. We have also improve the morphological terminology in the species description and discussion.

Comments 2: [Typos and English language]

Response 2: We have revised all the manuscript to correct all typos and we have corrected some expressions and phrases.

Reviewer 4 Report

dear authors,

I have read your manuscript with interest, and it was a pleasure to read. 

The Xanthoparmelia genus is quite interesting, and will provide surely new interesting surprises in the future. I am especially quite interested in the presence of a taxon which is associated with the southern hemisphere clade in the Mediterranean area. 

As far as the manuscript is concerned, I observed several typos. As an example, in the abstract, three lines from the bottom, there is a double blank. It is not the only case in the manuscript. I also found at least one case of a double dot (..).

The citations in the text do not follow the guidelines ot Journal of Fungi, and should be corrected. 

As far as the English language is concerned, I found some small issues. As an example, in the Introduction, first sentence: “The diversity of extant fungi and the estimated enormous number is one of the great questions facing evolutionary biology”, but the word "extant", which I find superfluous, “the estimated” probably should be “their estimated”. The second sentence begins with “As has been frequently noted”, which should probably be “As it has…”. Please, review carefully the manuscript to address any problem with language and grammar.

I have only one question. In the Materials and Methods you wrote that "Two Xanthoparmelia specimens were used to generate new sequences". Why two, if you collected several specimens? Would it not be better to use more specimens?

Best regards

SM

See above

Author Response

Thank you very much for taking the time to review this manuscript and thank you very much for your comments which have helped to improve this manuscript. We will provide a point-by-point response:

Comments 1: [As far as the manuscript is concerned, I observed several typos. As an example, in the abstract, three lines from the bottom, there is a double blank. It is not the only case in the manuscript. I also found at least one case of a double dot (..).]

Response 1: Thank you for pointed this out. We have revised all the manuscript to correct all typos.

Comments 2: [The citations in the text do not follow the guidelines ot Journal of Fungi, and should be corrected]

Response 1: We will include the citations in the text numbered and placed in square brackets, following the recommendations for journal style in the revised manuscript.

Comments 3: [As far as the English language is concerned, I found some small issues. As an example, in the Introduction, first sentence: “The diversity of extant fungi and the estimated enormous number is one of the great questions facing evolutionary biology”, but the word "extant", which I find superfluous, “the estimated” probably should be “their estimated”. The second sentence begins with “As has been frequently noted”, which should probably be “As it has…”. Please, review carefully the manuscript to address any problem with language and grammar.]

Response 3Thank you for your comment. We have carefully reviewed the entire manuscript to improve the English language.

Comments 4: [In the Materials and Methods you wrote that "Two Xanthoparmelia specimens were used to generate new sequences". Why two, if you collected several specimens? Would it not be better to use more specimens?]

Response 4: We agree that our study raises the possibility that the populations of the two localities may have significant differences, but as we wrote in line 505 this variability ‘falls beyond the scope of our current contribution’. To achieve our objective in this manuscript of reassessing the genus circunscription of Lecanora olivascens it was sufficient to analyse with molecular markers one sample from each locality.